Patellar tendon adaptations to resistance training in young women using combined oral contraceptives

Vesterhus Ingvild ingvild@vesterhus.net 1
Hesseberg Eirik R. 1
Fjeldberg Ken 1
Engstad Martin K. 1 2
Paulsen Gøran 1
Hansen Mette 3
Nordez Antoine 4 5
Lacourpaille Lilian 4
Seynnes Olivier R. 1
1 Department for Physical Performance, Norwegian School of Sport Sciences , Oslo , Norway
2 Department of Nutrution, University of Oslo , Oslo , Norway
3 Department for Public Health, Aarhus University , Aarhus , Denmark
4 Mouvement - Interactions - Performance, Université de Nantes , Nantes , France
5 Institut Universitaire de France , Paris , France
Zech Astrid
Electronic publication date: 2025 Jun 12
Publication date: 2025
Volume: 13
Electronic Location ID: e19581
Received 2024 Dec 11; Accepted 2025 May 19
Copyright: ©2025 Vesterhus et al.
Copyright year: 2025
Copyright holder: Vesterhus et al.
License: This is an open access article distributed under the terms of the Creative Commons Attribution License, which permits unrestricted use, distribution, reproduction and adaptation in any medium and for any purpose provided that it is properly attributed. For attribution, the original author(s), title, publication source (PeerJ) and either DOI or URL of the article must be cited.
License URL: https://creativecommons.org/licenses/by/4.0/

Keywords: Oestrogen, Tendon hypertrophy, Tendon stiffness, Hormonal influence

Funding: The authors received no funding for this work.

==============================
Background

The study aimed to examine the impact of combined oral contraceptive pill (OC) use on patellar tendon (PT) adaptation to resistance training in young women.

Methods

Fifteen users of OC (28 ± 3 years) (OC group) and 17 eumenorrheic non-users (32 ± 5 years) (NOC group) performed heavy resistance training of the knee extensors over a period equivalent to three menstrual or pill cycles. Maximal isometric strength of the knee extensor muscles, PT cross-sectional area (CSA), tensile stiffness, and shear wave velocity (SWV) were measured before and after the intervention using combined ultrasonography and dynamometry.

Results

The training period increased maximal isometric strength in both groups (≈11%, P < 0.001) with no significant interaction with OC use (p = 0.965). Likewise, a small yet significant increase in proximal tendon CSA was observed (1.5 ± 1.6% for both groups, main training effect P < 0.001) without any significant interaction with OC use (p = 0.267). Tendon tensile stiffness also increased significantly (18.9 ± 26.3% in the OC group and 28.2 ± 35.1% in the NOC group, main effect: P < 0.001) but was not significantly affected by OC use (interaction effect: p = 0.428). Tendon SWV measurements yielded similar results, indicating a main effect of training (+12% on average, p = 0.024) but no significant interaction with OC use.

Conclusion

These findings suggest that OC use does not affect the increase in PT CSA and mechanical properties following short-term resistance training in young untrained females.

Introduction

Combined oral contraceptives (OC) are commonly used by over 150 million women globally, for menstrual regulation and contraception (UN. Population Division, 2022). OC contain synthetic oestrogen and progesterone, leading to daily fluctuations in these hormones after each pill intake, and a significant reduction in the body’s natural oestrogen and progesterone levels (Cooper & Patel, 2024). Oestrogen and progesterone receptors are present in tendons and ligaments (Faryniarz et al., 2006), suggesting a potentially important role of female steroid hormones in the maintenance of connective tissue. In vitro studies have shown that exposure to oestrogen in human cruciate ligament cell cultures appears to inhibit fibroblast proliferation and the expression of procollagen type I (Yu et al., 1999), while this inhibitory effect is attenuated when oestrogen is combined with progesterone (Yu et al., 2001). Another study on engineered ligaments found that exposure to oestrogen led to a decrease in lysyl oxidase–an enzyme crucial for maintaining collagen cross-links–and a reduction in mechanical stiffness (Lee et al., 2015). The effects of combined OC are challenging to predict based on in vitro findings alone, likely due to the complexity of hormonal interactions and the varying influences of endogenous and exogenous hormones. Therefore, it is essential to investigate the potential impact of OC further (Hansen & Kjaer, 2016).

The essential role of tendons in the locomotor apparatus (Holt & Mayfield, 2023), and their potential as proxies for estimating changes in other connective tissues, make it highly relevant to study the influence of OC use on adaptations to training. In that respect, recent research has particularly focused on 2nd and 3rd generation OC, due to their popularity and their potential effect on mitigating certain circulating growth factors involved in collagen synthesis (Hansen et al., 2009b). Studies in young women using microdialysis and measurements of collagen fractional synthesis rates suggest that regular use of 2nd and 3rd generation OC reduces collagen synthesis both at rest (Hansen et al., 2009b) and in response to exercise (Hansen et al., 2008), compared to non-use of hormonal contraception. Despite this evidence, the impact of OC on tendon properties remains unclear. When comparing OC users to non-users, some studies have reported greater strain in the tendons of OC users (Bryant et al., 2008; Morse et al., 2013). However, no differences in tendon stiffness and cross-sectional area (CSA) have been found between OC users and non-users, either among athletes (Hansen et al., 2013; Hicks et al., 2017) or following a training intervention (Dalgaard et al., 2019).

The discrepancies in findings among studies exploring the influence of OC on tendons may be attributed to several methodological considerations. While strain measurements characterize tendon behaviour under standardized loading conditions, they do not capture the non-linear behaviour of the loaded tendon or how the tendon resists deformation (i.e., stiffness), which is a fundamental property of tendon function. Furthermore, many studies inadequately control the menstrual cycle phase in non-users and often rely on self-reported menstrual cycle phase information (Elliott-Sale et al., 2021). This increases the likelihood of non-users being tested in different phases of the menstrual cycle, which may affect measurements of tendon properties due to fluctuations in oestrogen (Hansen et al., 2009a; Lee et al., 2015). Similarly, consistent timing of tests during either the active or inactive pill phase in OC users may reduce the risk of bias in repeated testing. Consequently, it is advisable to standardise the testing process with respect to the menstrual cycle and pill phases. Additionally, many previous studies lack crucial information regarding the type of OC, including the dosage of ethinyl oestradiol and the type and dose of progestin. Oral contraceptives can contain varying amounts of ethinyl oestradiol or, in some cases, none at all. Moreover, the progestins included in OC vary significantly in their potency and androgenic properties, directly influencing their anabolic potential. For example, some progestins exhibit high androgenicity, leading to more pronounced anabolic effects, while others possess antiandrogenic properties that can mitigate these effects (Burrows & Peters, 2007). Therefore, studies on the effects of OC should at a minimum provide detailed information on the type and dosage of synthetic hormones.

To date, only one intervention study has explored the potential influence of combined OC (3rd generation) on tendon adaptation to resistance training (Dalgaard et al., 2019). This study found no significant effect of OC on changes in tendon cross-sectional area (CSA) in response to training. However, it did not report on the influence of OC on training-induced changes in tendon stiffness, leaving open the possibility that a small effect of OC may have gone undetected. To our knowledge, no previous studies have investigated the combined effects of OC and resistance training on tendon stiffness. Therefore, a longitudinal study examining training adaptations in both tendon morphology and mechanical properties could provide comprehensive evidence of the influence of OC use. The purpose of the present study was to measure the properties of the patellar tendon (PT) in women who either use combined OC or do not use any hormonal contraception, over a duration equivalent to three menstrual, or pill, cycles. Based on cellular-level findings, we hypothesized that the training-induced increase in tendon stiffness and CSA would be reduced in OC users.

Materials and Methods

Participants

Thirty-nine untrained women were recruited and assigned to two groups based on OC usage: non-users of OC (NOC group; n = 19) or OC users (OC group; n = 20). Participants had to be between 18 and 40 years old and not engaged in strength training (fewer than one session per week in the last 6 months). Non-users of OC were required to not have used OC for at least 6 months prior to the study and to have maintained a regular menstrual cycle ranging from 24 to 32 days before the intervention. OC users were included if they had been using combined OC pills of the 2nd or 3rd generations for at least 6 months prior to the intervention and had not used any other form of hormonal contraception. These types of OC were targeted because, unlike progestogen-only pills, they all contain both oestrogen and progestins with androgenic action, despite variations in the oestrogen content and progestin potency/androgenicity. Menstrual and pill cycles were monitored using a phone app (Clue, Biowink, GmbH) and ovulation kit tests (Babyplan Ovulation test stick, BPL Diagnostics As, Oslo, Norway). Exclusion criteria included orthopaedic or metabolic disorders affecting participation, current pregnancy, or use of progestogen-only (“mini”) pills. All participants provided written informed consent and were thoroughly informed about the study’s risks and benefits. The study protocol adhered to the Declaration of Helsinki and received approval from the Norwegian School of Sports Sciences ethics committee (IRB approval number 220-170322).

Experimental design

The study aimed to compare the effects of resistance training on PT between the NOC and OC groups over three menstrual or pill cycles, respectively. Each participant attended the laboratory for a familiarization session approximately four days prior to baseline testing. This was done to reduce any potential learning effects of maximal strength tests and to train participants to develop consistent contraction patterns for the tendon stiffness tests (Fig. 1). Testing was performed on the right leg and included measurements of maximal knee extensor isometric moment, PT CSA, tensile stiffness, and shear wave velocity (SWV). All ultrasound scans requiring manual analysis were anonymised. Body composition was assessed by Dual-Energy X-ray absorptiometry (DXA) (Lunar iDXA, Healthcare, Madison, WI, USA). Additionally, blood samples were drawn during the training intervention to control for differences in hormonal levels between groups. All measurements were repeated after the training period. DXA, strength measurements, and ultrasonography of the PT were conducted at least 2 days after the last training session. To control for potential hormonal influence at the time of testing, the NOC group was tested 3–9 days after the first day of menstrual bleeding, while the OC group was tested on days 3–9 after the last pill. During these days, both groups exhibited comparatively low endogenous levels of oestrogen and progesterone. Additionally, OC users have negligible levels of synthetic hormones, as the half-life of these hormones is reported to be less than 24 h (Goldzieher & Stanczyk, 2008). Therefore, we aimed to test the accumulated effect of OC use or non-use on the measures of muscle strength, and tendon properties and reduce the bias from differential levels of endogenous and exogenous hormones on the days of testing. Training sessions were conducted twice a week. Participants were instructed to maintain their habitual dietary intake and daily physical activity levels throughout the intervention period.

Figure 1 Study overview.

FP, follicular phase. WP, withdrawal phase. PP, pill phase. LP, luteal phase. ↓ represents the time-point of testing/training during the training intervention.

Resistance training program

This study was part of a broader investigation (unpublished at the time of submission of this article) into the impact of OC on training outcomes, which included various muscle groups and resistance training methods. The training segment relevant to this study focused on the quadriceps and included leg press and knee extensions exercises. Training sessions were supervised by experienced personal trainers during the first two sessions, and subsequently every other week. Each training session began with a 5-minute warm-up. Training volume progressively increased across three cycles: the first cycle consisted of 2–3 sets of 8–12 reps, the second cycle of 2–4 sets of 6–10 reps, and the third cycle of 3–4 sets of 4–6 reps. Training intensity was adjusted using the principles of repetitions in reserve (RIR) and rate of perceived exertion (RPE). Resistance training using RIR was chosen since it is an effective method for autoregulation, allowing individual adjustments based on daily readiness and performance levels (Larsen, Kristiansen & Van den Tillaar, 2021). The intensity increased progressively across each cycle: week one with RPE 7/RIR 3, week two with RPE 8/RIR 2, week three with RPE 9/RIR 1, and week four with RPE 10/RIR 0. Participants were instructed to interspace training sessions for at least 48 h.

To standardize the training duration, the OC group was instructed to complete a three-week pill consumption phase (PP) and a one-week pill withdrawal phase (WP), to match a typical 28-day menstrual cycle. Training frequency for the NOC participants was adjusted to match the total number of training sessions across all participants, totalling 24 sessions over three menstrual/pill cycles, as training frequency has been reported not to influence strength gain or muscle mass when training volume is matched (Grgic et al., 2018). Training frequency was reduced for women with longer menstrual cycles (>28 days), while it was increased for women with a shorter cycle (<28 days). If participants missed a training session due to illness or other reasons, the session was postponed to the following week.

Blood sampling

Blood was drawn from an antecubital vein in the morning after a 12-h fast, during the early follicular phase (FP) and mid-luteal phase (LP) for the NOC group, and during the withdrawal phase (WP) and pill phase (PP) for the OC group, in the first and third cycles of training (Fig. 1). Serum levels of oestradiol, progesterone, follicle-stimulating hormone (FSH), and luteinizing hormone (LH) were measured, and the mean value of the two cycles was computed for each time point.

Isometric dynamometer strength

The maximal isometric moment of the knee extensor muscles was measured using an isometric dynamometer (Humac Norm; Computer Sports Medicine Inc., Stoughton, MA, USA) with participants seated and the hip angle at 70°. The participants performed three maximal voluntary isometric contractions (MVC) of the knee extensors at a knee angle of 60°. Each trial was separated by at least two minutes of rest. If the highest MVC differed from the second highest by more than 10%, up to two additional attempts were made to ensure the highest MVC was achieved for each strength test. The average of the two attempts with the highest force was used for further analysis.

Patellar tendon morphology

Tendon morphology was assessed using B-mode ultrasonography (Mach 30; Hologic SuperSonic Imagine, Aix-en-Provence, France) with a 50-mm transducer (18-5 MHz). Tendon length was measured externally by determining the distance between the patella apex and the PT insertion on the tibial tuberosity. The PT CSA was measured at three sites: (1) just distal to the PT insertion; (2) at PT mid-length and (3) at the level of the tibial plateau (∼75% of tendon length distally). Rather than capturing standard static images at the target locations, we aimed to enhance scan position matching and between-session reliability by recording scan videos over the target areas and matching specific frames. Two scan videos were collected for each target location while sliding the transducer in the proximo-distal direction over approximately one cm. Matching still frames from the ultrasonography videos taken at baseline and post-intervention were identified offline (Horos v3.3.6; Horos Project) based on anatomical landmarks (e.g., blood vessels, connective tissue within adipose tissue, etc.). Finally, PT CSA was outlined manually in the matched frames (Fig. 2). The mean CSA was calculated as the average of the CSA outlined at the three target locations. Tendon CSA was measured during familiarisation and baseline sessions in a sub-sample of 13 participants, indicating excellent (Koo & Li, 2016) inter-day reliability, with a 95% intraclass correlation coefficient (ICC (3, 1)) of 0.99 for all three tendon regions, and a raw typical error (TE) of 0.9 mm2, 0.6 mm2, and 0.9 mm2, respectively, for the proximal, middle and distal regions. The corresponding coefficient of variation (CV) values for this measurement were 0.5%, 0.4%, and 0.4%, respectively, for the proximal, middle, and distal regions.

Figure 2 Outline of patellar tendon cross-sectional area for one participant at baseline (left) and post-training (right).

White arrows point at examples of landmark references that were used to match frames between time points.

Patella tendon stiffness

Tendon stiffness was measured using ultrasonography. The participants were seated in a custom-made, rigid ergometer (GYM2000, Geithus, Norway). A leg cuff connected to a strain gauge (U2A, Hottinger Baldwin Messtechnik GmBh, Darmstadt, Germany) through a rigid steel rod perpendicular to the lower leg was mounted just above the medial malleolus. A 50-mm ultrasonography probe (L-18,5) was fitted into an adjustable cast (a custom-made patella-version of the ProbeFix Dynamic prototype from Usono, Eindhoven, The Netherlands) accommodating an acoustic gel pad, (Aquaflex Gelpad, Parker Laboratories Inc., Fairfield, NJ, USA) and secured to the knee with elastic bands. The ultrasonography probe and cast were adjusted to display the PT sagittally, including the patella apex and the tibial plateau throughout the isometric contractions. The raw force signal was sampled (MP150; Biopac, Goleta, CA, USA) at 1,000 Hz. The computer and ultrasound machine were interconnected to allow synchronous sampling of all data using a custom-made trigger device.

Participants performed three consecutive contractions at 80% MVC to precondition the tendon, followed immediately by a gradually increasing force until MVC, over a 2–3 s period. Participants were verbally encouraged during contractions, given visual and oral feedback on their performance, and instructed to follow a loading curve displayed on a computer screen to ensure a constant increase in muscle torque (∼70 Nm/s). They repeated this test until they achieved four successful attempts, as subjectively assessed from the linearity of the torque trace, the torque peak (within 20% variation), and satisfactory ultrasound scans. Each trial was separated by a 2–3-minute rest period. Routine modifications to the knee extensor moment (e.g., adjusting for co-contraction) were deliberately omitted from this protocol to avoid reducing sensitivity or due to logistical constraints (further details can be found in the limitations section).

PT stiffness was calculated from the processing of force and ultrasound data (Python, v. 3.10). Raw force data were first processed with a lowpass filter with a 30 Hz cut-off. Then the knee extension moment was calculated by multiplying the sampled force by the external moment arm. The external moment arm was measured using a measuring band from the leg cuff (mid-width) to the rotational axis of the knee joint. Quadriceps force was obtained by dividing the measured knee extension moment by the internal moment arm. The latter was measured from close-up photographs of the lateral side of the knee, as the perpendicular distance between the PT frontal edge and the knee lateral epicondyle (considered the rotational axis) using ImageJ/Fiji (Schindelin et al., 2012). The location of the lateral epicondyle was estimated by palpation and marked on the skin before taking the photographs. An average value from two analyses of the same photograph was used. The same estimated internal moment arm was used at baseline and post-training, since this parameter is not expected to change over time. Tendon elongation was measured as the frame-to-frame difference in distance between the patella apex and the tibial plateau, using a custom implementation of the Lucas–Kanade method for optical flow calculation (Python, v. 3.10). The tendon elongation data were plotted against the tendon force data. The force–elongation curves from the trials of each individual were resampled to 100 points and fitted with a 2nd-degree polynomial. Trial curves with a coefficient of determination lower than 0.90 were excluded. The remaining curves were then standardized to a common force level, defined as the lowest peak force observed either pre- or post-training, and subsequently averaged to a mean force–elongation curve. Tendon stiffness was calculated as the slope of this mean curve within the range of 60% to 100% of the maximal force (Werkhausen et al., 2018; Eriksen et al., 2019). Tendon stiffness was measured from familiarisation and baseline sessions in a sub-sample of 13 participants, and indicated good (Koo & Li, 2016) inter-day reliability, with an ICC (3, 1) of 0.88, a TE of 501 N/mm, and a CV of 7.6%.

Tendon shear wave velocity

Tendon shear wave velocity (SWV) was measured sagittally with the same ultrasound apparatus, and the probe was manually held on the frontal plane of the lateral and the medial side of the tendon. SWV is related to the shear modulus of biological tissues (Hug et al., 2015) and can be used as a proxy for the tissue’s Young’s modulus. The scans were performed with the participants seated and relaxed on the dynamometer chair, with their knee joint at a 90° angle. Since tendon tissue is very stiff, SWV values can reach the maximal measurement capacity of the machine in certain areas. To avoid any bias due to saturation, recordings were discarded if more than 5% of pixels (averaged across frames) indicated a velocity greater than 90% of the maximal measurable velocity (20 m/s). All scans were analysed using an automated script (Python v. 3.10, swepy, Seynnes, 2023). Tendon SWV was measured from familiarisation and baseline sessions in a sub-sample of 12 participants, and indicated moderate to excellent (Koo & Li, 2016) inter-day reliability, with ICCs (3, 1) of 0.60 (medially), 0.94 (laterally) and 0.90 (mean of both sides), and a TE ranging 3.7 m/s (medially), 2.2 m/s (laterally), 2.1 m/s (mean). The corresponding CV values for this measurement were 16.8%, 9.9% and 10.8%, respectively, for the medial side, lateral side and for the mean of both sides.

Statistical analysis

Differences between groups at baseline were tested with a Student’s t-test. Since most variables related to blood analysis were non-normally distributed (Shapiro–Wilk’s test), baseline differences between cycle and between groups were tested with a Wilcoxon signed-rank test and a Mann–Whitney U test, respectively. For all other variables, a repeated mixed-factor ANOVA was used to test whether the training intervention resulted in significant changes and whether there was an interaction with OC use. For most variables a two-way ANOVA (training x group) was used, while a three-way ANOVA (training x group x location) was used for tendon CSA, to assess interactions with tendon sites of measurement. Assumptions of normality and sphericity were verified for all ANOVA analyses. Post hoc tests with Bonferroni correction were carried out where appropriate. The Pearson’s product moment correlation coefficient was also calculated to investigate relationships between variables of interest. Analyses were performed in JASP 0.19.0 (JASP Team, 2024). Data are reported as mean ± standard deviation. Differences in main outcome variables are reported as mean and 95% confidence interval.

Results

Participants

Six participants dropped out from the training intervention for reasons not directly related to the training intervention, or were discarded because of a long menstrual cycle duration (44 days, n = 1) (Janse de Jonge, Thompson & Han, 2019). Ultimately, 32 women completed the study, 17 women in the NOC group and 15 women in the OC group. The broader investigation connected to the present study focused on the influence of OC on adaptations to resistance training. Therefore, the target sample size was based on an a priori power calculation from the changes in muscle CSA induced by conventional resistance training in OC users and non-users, as previously reported by Dalgaard et al. (2019), which recommended 16 participants per group. While the relationship between increases in muscle size and tendon properties is indirect, several studies (Kongsgaard et al., 2007; Seynnes et al., 2009; Centner et al., 2021) demonstrating changes in PT stiffness and CSA with training had a similar or lower sample size. A posteriori power calculation indicate that the statistical power was sufficient for tendon mean CSA (β > 0.99) but may have been insufficient for tendon stiffness (β = 0.71) and was insufficient for the mean SWV (β = 0.24). Women in the OC group had used OC for at least 6 months prior to the study (mean duration: 8.1 ± 4.6 years), while women in the NOC group had abstained from OC usage for at least 6 months prior to the intervention (mean duration: 6.2 ± 6.4 years). The OC participants used 2nd or 3rd generation contraceptives, with the following dosages of synthetic oestradiol and progestin: (1) 30 µg ethinyl-oestradiol and 150 µg levonorgestrel (n = 11); (2) 20 µg ethinyl-oestradiol and 100 µg levonorgestrel (2nd generation OC) (n = 2); (3) 30 µg ethinyl-oestradiol and 150 µg desogestrel (3rd generation) (n = 1); (4) 20 µg ethinyl-oestradiol and 150 µg desogestrel (3rd generation) (n = 1). To ensure regular ovulatory function in the NOC group, ovulation was verified using both ovulation kit tests and luteal phase serum progesterone levels (≥16 nmol/L considered indicative of ovulation (Janse de Jonge, Thompson & Han, 2019)). Four participants had slightly lower progesterone concentrations (5–13 nmol/L), possibly due to timing of sampling. However, ovulation was confirmed by positive test kits in all cycles for two of them, and in two out of three cycles for the remaining two. Thus, all participants in the NOC group were considered to have ovulated during the test cycle and classified as eumenorrheic. There was no significant difference between groups at baseline in height, weight, BMI, body fat, and lean mass, but a 3-year age difference was found (Table 1). A sub-analysis (not included for clarity) was conducted for all variables, focusing only on participants using 2nd generation OC. This analysis did not yield any results different from those described below. Training compliance was not statistically different between groups (23.5 sessions completed in both groups, from a total of 24 planned sessions, p = 0.868).

Table 1 Participants’ characteristics.

	OC group (n = 15)	NOC group (n = 17)	p-value	
Age (years)	27.9 ±2.8	31.6 ±4.8	0.014	
Height (cm)	171.4 ±5.6	170.2 ±8.0	0.616	
Weight (kg)	70.0 ±14.7	69.3 ±9.0	0.871	
BMI (kg/m2)	23.8 ±4.7	24.0 ±2.9	0.898	
FM (kg)	26.4 ±10.8	20.6 ±7.1	0.080	
FFM (kg)	44.3 ±5.3	44.4 ±4.2	0.980	
Notes.

OC group combined oral contraceptive users

NOC group oral contraceptive non-users

BMI body mass index

FM fat mass

FFM fat-free mass

Data are reported as mean ± standard deviation.

Blood hormonal profile

The concentrations of sex hormones confirmed the expected difference between the OC and NOC groups (Table 2). Oestradiol and FSH concentrations were higher in the NOC group post-menstruation (FP) and post-ovulation (LP) compared to the OC group during withdrawal phase (WP) and pill phase (PP). Progesterone and LH concentrations showed no significant difference between FP (NOC) and WP (OC) but were higher in LP (NOC) compared to PP (OC). Progesterone was undetectable (detection level = 3 nmol/L) in most NOC participants in FP and all OC users in WP and PP. LH values showed no significant difference between phases within groups. In the NOC group, oestradiol, progesterone, and sex hormone binding globulin (SHBG) were higher post-ovulation (LP) than post-menstruation (FP), with FSH higher in FP than LP. In the OC group, oestradiol and FSH were higher during the weeks of pill usage (PP) than the non-pill week (WP), but SHBG was higher in WP.

Table 2 Serum sex hormones.

	OC group (n = 15)	NOC group (n = 17)	
	PP	WP	Cycle mean	FP	LP	Cycle mean	
Oestradiol (nmol/L)	0.09 ±0.04	0.07 ±0.01*	0.08 ±0.02†	0.35 ±0.30	0.52 ±0.22*	0.44 ±0.19	
Progesterone (nmol/L)	<3.0	<3.0	n/a	<3.0	27.7 ±13.2*	n/a	
FSH (IU/L)	5.6 ±3.7	2.8 ±2.5*	4.2 ±2.6†	8.4 ±2.8	4.8 ±1.7*	6.6 ±1.9	
LH (IU/L)	2.7 ±1.8	2.5 ±3.2	2.6 ±2.0†	7.9 ±10.0	5.4 ±3.0	6.6 ±5.1	
SHBG (nmol/L)	110.6 ±33.1	122.8 ±34.1*	116.7 ±32.7†	54.9 ±20.6	59.7 ±21.8*	57.3 ±21.1	
Notes.

OC group combined oral contraceptive users

NOC group oral contraceptives non-users

FSH Follicle-stimulating hormone

LH luteinizing hormone

SHBG sex hormone binding globulin

Data are reported as mean ± standard deviation.

* p < 0.05 when comparing to the other phase within group.

† p < 0.05 when comparing between groups.

Maximum isometric knee extension strength

No difference in maximum isometric knee extension strength was observed between the groups at baseline (p = 0.975). There was a main effect of training, as maximum isometric knee extension strength increased in both groups (mean difference: +21.7 Nm, 95% CI [16.0–27.3]) following the training intervention, from 212.8 ± 33.0 Nm to 234.5 ± 30.8 Nm in the OC group and from 213.2 ± 33.9 Nm to 234.6 ± 33.7 Nm in the NOC group, p < 0.001, ηp2=0.653. However, no interaction effect between OC use and training was observed (p = 0.956, ηp2=1.045 × 10−4).

Patella tendon CSA

Ambiguous tendon borders made it impossible to evaluate the tendon CSA in two participants of the NOC group. Therefore, PT CSA was examined in 15 women from the NOC group, and 15 from the OC group. No significant difference was observed between the groups at baseline for tendon CSA at various sites (proximal, mid, distal), or for the mean of all sites (0.109 < p < 0.190). The results showed a significant main effect of training (mean difference: +0.73 mm2, 95% CI [0.46–0.99], p < 0.001, ηp2=7.500 × 10−4) and a training × location interaction effect (p < 0.049, ηp2=1.300 × 10−4, Fig. 3). This indicates that, for the combined groups, the training increased tendon CSA , but the increase was not uniformly along the tendon length. Post hoc comparisons showed that the tendon CSA increased only at the proximal site (mean difference: +1.10 mm2, 95% CI [0.38–1.83], p < 0.001, 1.5 ± 1.6%, Fig. 4). No interaction was found between the training effect and OC use (p = 0.267, ηp2=3.264 × 10−5, Fig. 3). No significant correlation was found between changes in CSA following resistance training and baseline PT CSA or mean serum oestradiol levels.

Figure 3 Changes in patellar tendon CSA in OC group (n = 15) and NOC group (n = 15) at 3 sites along the length of the patellar tendon.

(A) Combined mean tendon CSA before (PRE) and after (POST) the training period. A significant main effect of training (p < 0.001) and a significant training x location interaction effect (p < 0.05) were observed. (B) Changes in tendon CSA from baseline to post training. PT CSA: patellar tendon cross-sectional area, Prox: proximal, Mid: mid-portion, Dist: distal, OC: oral contraceptive users, NOC: oral contraceptive non users. *Significant main effect of training at a given location shown from post hoc analysis (p <  0.001). Data are reported as mean ± standard deviation.

Figure 4 Representative force–elongation curves from one participant, before (Pre) and after (Post) the training period.

The resultant curve from all trials (scatter points) is depicted as a dark blue curve. Tendon stiffness was calculated as the slope of this curve between 60% and 100% of the maximal force at baseline or post training, whichever one was the lowest. Data are reported as mean ± standard deviation.

Patella tendon stiffness

The fitting of the force–elongation curve failed to meet the inclusion criteria for two participants from each group, making it impossible to reliably assess tendon mechanical properties. As a result, the analysis included 15 NOC and 13 OC participants. Patellar tendon moment arm length did not differ between groups (35.5 ± 2.8 mm for the OC group and 37.1 ± 4.4 mm in the NOC group, p = 0.262). No difference in tendon stiffness was detected at baseline (p < 0.583). Training had a main effect on PT stiffness as measured with ultrasonography during isometric contractions (mean difference: +647 N/mm, 95% CI [280–1014], p < 0.001, ηp2=0.336) (Fig. 5). In the OC group, PT stiffness increased by 18.9 ± 26.3%, while in the NOC group, PT stiffness increased by 28.2 ± 35.1% (Fig. 5). However, no significant effect of OC use on the change in PT stiffness was observed (interaction, p = 0.428, ηp2=0.024). Additionally, no correlation was observed between baseline tendon stiffness or mean serum oestradiol and the change in tendon stiffness during the resistance training period.

Figure 5 Changes in patella tendon stiffness.

Patellar tendon stiffness in oral contraceptive users (OC, n = 13) and non-users (NOC, n = 15), before (PRE) and after (POST) training. PT: patellar tendon. *Significant main effect of training (p < 0.001). Data are reported as mean ± standard deviation.

Patella tendon shear wave velocity

Following the exclusion of trials with saturated pixels (see ‘Methods’), the analysis of PT SWV included 11 OC and 15 NOC participants. There were no baseline SWV differences between the groups on either side (medial, lateral) or for the mean of both sides (Table 3). Training had a main effect on the mean SWV of both sides (mean difference: +1.29 m/s, 95% CI [0.19–2.39], p = 0.024) and the SWV on the medial side separately (mean difference: +1.04 m/s, 95% CI [0.31–1.78], p = 0.007), but not on the lateral side (p = 0.236, Table 3). However, no interaction effect was found between OC usage and training (p = 0.405–1.000). Also, no correlation was observed between baseline SWV and the changes in SWV with resistance training at either the medial site, lateral site, or mean (r = −0.25–0.14, p = 0.270–0.718).

Table 3 Shear wave velocity.

	OC group	NOC group	Training main effect	Group × training interaction	
	n	Pre	Post	n	Pre	Post	p-values	ηp2	p-values	ηp2	
Lateral side SWV (m/s)	11	6.8 ± 2.2	7.5 ±3.9	15	7.3 ±2.7	7.9 ±2.6	0.236	0.058	1.000	1.396 × 10−8	
Medial side SWV (m/s)	11	8.3 ± 2.8	9.0 ±3.9	15	6.8 ±2.9	8.2 ± 3.9	0.007	0.263	0.405	0.029	
Mean SWV (m/s)	11	7.6 ±1.9	8.2 ± 3.0	15	7.1 ± 2.4	8.0 ± 2.8	0.024	0.195	0.747	0.004	
Notes.

SWV shear wave velocity

OC group combined oral contraceptive users

NOC group oral contraceptive non-users

Data are reported as mean ± standard deviation.

Discussion

This study aimed to explore how OC use affected adaptations of the PT in women undergoing resistance training over the course of three menstrual or pill cycles. Training resulted in increased tendon CSA, stiffness, and SWV. However, OC use did not appear to influence the adaptations in tendon CSA or mechanical properties, as no interaction effect between OC use and training was observed. Therefore, our data do not support the hypothesis that OC use significantly inhibits tendon adaptations during a relatively short-term resistance training period.

Tendon hypertrophy

Training significantly increased tendon CSA, which is congruent with previous studies investigating PT morphological adaptation to short-term resistance training in males (Kongsgaard et al., 2007; Seynnes et al., 2009) and females (Vikmoen et al., 2016; McMahon et al., 2018; Dalgaard et al., 2019), although the increase was somewhat smaller in magnitude (≈1.5% proximally) than previously found in similar populations (Vikmoen et al., 2016; >6%, Dalgaard et al., 2019). This discrepancy could be due to differences in training stimuli and/or assessment methods (Wiesinger et al., 2015), such as ultrasonography versus MRI. However, training resulted in an increase in the muscle strength of the knee extensors, suggesting that the training program provided a sufficient stimulus for training adaptations.

The training-induced tendon hypertrophy did not significantly differ between the OC and NOC groups. This aligns with a previous training study involving users of 3rd generation OC and controls (Dalgaard et al., 2019). Previous observations suggested that OC users might have reduced tendon adaptation due to lower PT collagen synthesis rates compared to non-users, both at rest and in response to a single exercise bout. This may be explained by a reduced bioavailability of insulin-like growth factor 1 (IGF-1) in combined OC users (2nd and 3rd generation) compared to non-users (Hansen et al., 2009b). IGF-I, which is expressed in tendons, has a documented stimulating effect on cell proliferation and/or collagen synthesis in both animal (Abrahamsson, Lundborg & Lohmander, 1991) and human models (Vestergaard et al., 2012). However, our results do not support the hypothesis of impaired training-induced tendon hypertrophy in OC users.

One possible explanation is that the influence of OC consumption under the present training conditions was not substantial enough to induce a difference in adaptation to training at the macroscopic level (tendon CSA). Another potential explanation is that the use of OC not only reduces the rate of tendon collagen synthesis but also decreases the rate of tendon collagen breakdown. If true, there might be no net difference in tendon collagen turnover and tendon CSA between OC users and non-users. This theory is supported by the fact that the tendon CSA was not different at baseline between groups in the current study, or in previous cross-sectional studies involving both athletes (Hansen et al., 2013) and non-athletes (Dalgaard et al., 2019). This is despite reports of a lower tendon collagen synthesis rate at rest, and in response to acute exercise (Hansen et al., 2009b). An overall lower tendon collagen synthesis rate in OC users compared to non-users could potentially lead to differences in tendon biomechanical properties (Kjaer et al., 2005) but this was not observed in the current study.

The inconsistency with cellular/molecular data may be related to the complexity of the changes caused by OC consumption. On the one hand, OC increases SHBG levels, which lowers free testosterone (Zimmerman et al., 2014) and may impact collagen synthesis negatively. On the other hand, increased cortisol levels resulting from OC consumption (Eisenhofer et al., 2017) might upregulate collagen metabolism. Additionally, despite lower endogenous oestrogen and progesterone levels in OC users (Table 2), the overall exposure to and effect of bioactive hormones (both endogenous and exogenous) in OC users may not differ significantly from non-users over the course of the training period. This is especially considering the more potent effect of ethinyl oestradiol over endogenous 17β-estradiol (Coelingh Bennink, 2004).

Changes in tendon biomechanical properties

Resistance training has consistently been observed to increase tendon stiffness in men (Bohm, Mersmann & Arampatzis, 2015; Wiesinger et al., 2015) and in older women (Kubo et al., 2003). Additionally, PT stiffness and Young’s modulus are higher in the jumping leg than in the contralateral leg in female handball players (Hansen et al., 2013), supporting the notion that physical loading influences tendon biomechanical properties. We observed an increase in tendon stiffness after the training period (+19% in OC users and +28% in non-users). However, this increase was not statistically dependent on OC consumption, contrary to our initial hypothesis. This finding reflects the training effect observed for the SWV measured in the medial side of the PT (and for the average of the medial and lateral sides), where no interaction effect with OC use was observed. The relationship between tendon mechanical properties inferred via SWE and tensile testing is currently unclear. Although the moduli measured with the two methods are correlated at very low tensile strain in the Achilles tendon (Haen et al., 2017; Mifsud et al., 2023), their relationship at higher force levels remains elusive. This is likely due to the different approaches of the techniques and their specific methodological limitations. Shear wave velocity was included in this study as a complementary method to estimate tendon material properties, aiming to avoid the inherent variability associated with tensile tests (e.g., repeated voluntary contractions), rather than as a direct surrogate for the tensile elastic modulus. The effect of resistance training on tendon SWV is not well-documented due to the limited number of studies and the variation in ultrasound equipment and methodologies used. One study found no impact of short-term resistance training (Mannarino, Da Matta & De Oliveira, 2019), while other research reported an acute increase in SWV following heavy resistance exercise (Heales et al., 2018) and higher SWV in elite athletes compared to physically active controls (Götschi et al., 2022). In the present study, SWV measurements were sensitive enough to detect the training-induced increase but do not indicate an influence of OC. Collectively, these results are contrary to our initial hypothesis but are consistent with previous cross-sectional studies (Hansen et al., 2013; Hicks et al., 2017), which found no significant difference in tendon stiffness between OC users and controls. However, others have reported lower Achilles tendon strain values in OC users compared to controls (Bryant et al., 2008; Morse et al., 2013).

One interpretation for the lack of difference between groups is that OC do not substantially influence tendon stiffness due to their complex action on collagen metabolism. An alternative explanation could be that OC does have a minor influence, but the duration of the training intervention in the present study was insufficient to induce measurable changes. However, this appears unlikely given the lack of difference between groups at baseline, suggesting that the combined long-term use of OC and daily loading did not already induce a difference between groups by this time point. A third possible explanation might be related to the sensitivity of tendon mechanical property measurements. While our tendon CSA measurements demonstrated excellent reliability (TE ranged 0.6–1.2% of mean values), tensile stiffness measurements are typically less reliable (TE = 17.8% here, relative to mean values). This degree of typical error, which was even greater for SWV measurements, might have been too high relative to the differences being examined. Therefore, the development of more sensitive tests of tendon mechanical properties may be required to ascertain that OC consumption affects changes in these variables with training.

Strengths and limitations

The present study adds valuable contributions to the understudied yet important area of research on the influence of OC use on tendon adaptations to short-term resistance training in young eumenorrheic women. Since the effect of OC on tendon adaptations to training was expected to be small, we designed the protocol to optimize the detection of changes in main outcome variables. The tensile stiffness measurements were based on multiple trials and a common force level (Seynnes et al., 2015), and we measured SWV as a complementary indicator of tendon stiffness independent of voluntary contractions. Detection of changes in tendon CSA was optimised with an original method designed to accurately matching pre- and post-training scans offline (see ‘Patellar Tendon Morphology’). In addition to these precautions, our approach integrated several methodological considerations specific to research on women (Elliott-Sale et al., 2021). The endogenous hormonal profile of the participants was measured to ascertain the hypothesised differences between groups and to adapt the timing of the training programme and testing sessions. Although we standardized testing timing during the menstrual or OC cycle to account for hormonal variations (McNulty et al., 2020), the 3–9-day window might have placed some participants in the early consumption or mid-follicular phase, potentially altering hormonal environments. This broader time window was necessary due to logistical constraints, including lab availability and participant schedules. Additionally, we initially aimed to only include OC users using one type of 2nd generation OC. Due to recruitment challenges, we had to include four participants using a different type of 2nd generation or a 3rd generation OC. Monophasic OC of the 2nd and 3rd generation contain different quantities of oestrogen and different types of progestin with various levels of androgenicity and potency. For this reason, the four participants who did not use the same OC as the other 11 in the OC group (see details in ‘Participants’) were not exposed to identical hormonal influences. Importantly, a secondary analysis excluding these participants did not reveal any significant differences in the outcome measures. However, the loss of data (see ‘Patella Tendon Stiffness’, and ‘Patella Tendon Shear Wave Velocity’) affected the sensitivity of tensile- and particularly SWE-based stiffness, as evidenced by a posteriori power calculations. Future studies should focus on improving the methods of these variables and/or recruiting a larger sample size to enhance the robustness and reliability of the findings. Addressing these limitations could provide more definitive insights into the relationships between hormonal profiles, OC usage, and tendon stiffness.

Our study did not randomize the use of OC, implying that non-training groups would be necessary to fully understand the impact of OC-start on muscle mass and strength. In contrast, a positive aspect of our implementation was the inclusion of regular OC users, who have already passed through transient side effects (e.g., hormonal and psychological) occurring when starting to use OC. Had OC consumption been initiated at the same time as the exercise intervention, we would not be able to distinguish between their effects without the inclusion of non-exercise groups.

During tendon stiffness tests, we chose to leave out the contribution of antagonist (hamstring and gastrocnemius) co-activation. Although co-contraction may account for 10–30% of the resultant knee joint moment (Aagaard et al., 2000), its estimation via electromyography is complex and may be inaccurate at times (Raiteri, Cresswell & Lichtwark, 2015). This may particularly be the case in untrained women with more subcutaneous fat in their lower body than trained counterparts or men, which is shown to decrease signal amplitude and increase crosstalk (Kuiken, Lowery & Stoykov, 2003). Additionally, we did not correct for joint angular rotation during the stiffness testing because of logistical reasons. However, we used an ergometer that was stiff, and we assumed that the error margin caused by joint rotation at baseline would be similar post-training, as we measured stiffness at the same individual force level.

Finally, the guiding effect inherent to shear waves travelling through thin and stiff structures such as tendons should be considered (Brum et al., 2014; Helfenstein-Didier et al., 2016). This effect implies that that SWV is influenced by tendon thickness and is therefore not entirely related to the shear modulus. However, the small increase in tendon CSA suggests that the tendon thickness was only marginally affected by training, and the guided wave propagation was likely similar pre- and post-training. Additionally, changes in thickness were consistent across groups. Taken together, these considerations suggest that the guided wave propagation did not influence the main results of the present study.

Conclusions

The present study indicates that, under the present experimental conditions, OC does not influence tendon adaptation to short-term resistance training. This finding is significant, as OC is widely used particularly among young female athletes. However, if OC attenuates the adaptive capacity of tendons and other connective tissues over the longer term, it could negatively impact performance and increase injury risk. Further research is therefore warranted, possibly exploring the effects of other types of OC, using more sensitive methods to measure tendon mechanical properties, and extending the duration of training.

We are grateful to Victor Donker and the Usono company for their assistance and expertise in the development of the probe holders used in this experiment.

Additional Information and Declarations

Competing Interests

Author Contributions

Human Ethics

Data Availability

The authors declare there are no competing interests.

Ingvild Vesterhus conceived and designed the experiments, performed the experiments, analyzed the data, prepared figures and/or tables, authored or reviewed drafts of the article, and approved the final draft.

Eirik R. Hesseberg conceived and designed the experiments, performed the experiments, authored or reviewed drafts of the article, and approved the final draft.

Ken Fjeldberg conceived and designed the experiments, performed the experiments, authored or reviewed drafts of the article, and approved the final draft.

Martin K. Engstad conceived and designed the experiments, performed the experiments, authored or reviewed drafts of the article, and approved the final draft.

Gøran Paulsen conceived and designed the experiments, authored or reviewed drafts of the article, and approved the final draft.

Mette Hansen conceived and designed the experiments, authored or reviewed drafts of the article, and approved the final draft.

Antoine Nordez conceived and designed the experiments, authored or reviewed drafts of the article, and approved the final draft.

Lilian Lacourpaille conceived and designed the experiments, authored or reviewed drafts of the article, and approved the final draft.

Olivier R. Seynnes conceived and designed the experiments, analyzed the data, prepared figures and/or tables, authored or reviewed drafts of the article, and approved the final draft.

The following information was supplied relating to ethical approvals (i.e., approving body and any reference numbers):

The Norwegian School of Sports Sciences (IRB approval number 220-170322).

The following information was supplied regarding data availability:

The data is available at Zenodo: Vesterhus, I., Hesseberg, E., Fjeldberg, K., Engstad, M., Paulsen, G., Hansen, M., Nordez, A., Lacourpaille, L., & Seynnes, O. (2024). OC and tendon adaptations to resistance training [Data set]. Zenodo. https://doi.org/10.5281/zenodo.14259356.

The code to SWE analysis is available at Zenodo: Seynnes, O. (2025). swepy (v1.0). Zenodo. https://doi.org/10.5281/zenodo.14999519.

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
