# Peer review of "Patellar tendon adaptations to resistance training in young women using combined oral contraceptives"

_PeerJ, doi:10.7717/peerj.19581_

## Round 0.1 · original submission · Major Revisions

The reviewers have made many valuable and detailed comments. Please address all of them in a comprehensive revision

Reviewer 1 ·

Basic reporting

Line 27 -9: This sentence in the abstract is confusing. “Tendon tensile stiffness also increased significantly (18.9 ± 26.3 % in the OC group and 28.2 ± 35.1 % in the NOC group, p < 0.001) but was not significantly affected by OC use (p = 0.428).” Is this not contrary? Surely if the first sentence is correct, the second is not. Do you mean there was a main effect of training, but no interaction?

Line 208: proximal, middle or distal edge of leg cuff?

Lines 367-71: Needs rephrasing for clarity (369+).

Line 414-5: This sentence is confusing and I’m not sure what is trying to be said – tendon stiffness would be greater for a hypertrophied tendon, all else being equal.

Line 415-7: Again, I’m not sure what is trying to be said with this sentence – that the two groups had been exposed to a NOC or OC environment for a longer period of time before the study?

Line 427: This study doesn’t need to be the ‘first’ to be of value (it is not a useful metric). You are underselling the uniqueness and robustness of the study by writing it. I would encourage you to remove this wording and instead emphasize on adding contributions to an understudied yet important area of research.

Experimental design

There is no rationale provided for your choice of OC users, nor is there information provided about the consequence of users choosing a combi vs mini-pill (i.e. what role does estrogen play in tissue homeostasis) and why is that important to control for in your study. I’d like to see a stronger emphasis on the physiology of sex hormones throughout the paper, and how they relate to your findings.

Do you have information on whether individuals in the NOC ovulated or not during their cycles? Anovulatory cycles are common and heavily impact LP hormone levels. https://doi.org/10.1210/jcem.83.12.5334. If you do have this data, please include and describe how it might influence your findings (or not).

Line 96: Is the intention of excluding users of the mini-pill a way of homogenizing your OC group? You have not provided a rationale of why you have chosen to focus on the combi pill – even though various generations/combinations were included in the OC group. Is it a specific interest in estrogen?

Line 113: 3 – 9 days. NOC may have significantly elevated estrogen by 9 days. OC are on the next week pill for 2 days already. I don’t agree that sex hormones would be low / negligible at 9 days into the cycle. Whether this would affect outcome measures however is

Line 420: I appreciate the acknowledgement of tendon stiffness variability. Perhaps the force range you had calculated this parameter over was not optimal; many researchers use values up to 90% MVC as the final 10% is more variable.

Line 433-5: Related to my point above about excluding mini-pill users, you have not detailed anywhere why you chose this group, or the importance of one hormone over another. You call this a compromise, but there is no attempt to tell the reader why [this is a compromise].

Line 439 – 448: I find this explanation adequate, but rather late to the story. Perhaps a brief mention of these omissions is warranted in the relevant methods section. E.g. routine modifications to knee extensor moment (e.g. adjusting for co-contraction) that were deemed systematic in nature were not completed for logistical reasons. Further details can be found in limitations section…

Validity of the findings

I commend the authors for their relevant study – there is little known about contraceptive use / hormone cycles on tendon properties.

I find the data and statistical tests acceptable and all conclusions are supported.

Additional comments

The study examines the effect of heavy resistance training on the mechanical and morphological properties of the patellar tendon. The study concludes that use of oral contraceptives does not affect the adaptive response of the patellar tendon to short-term training in young untrained females.

I found the discussion to be lacking critique and detail in places, which can make it seem less comprehensive or lacking depth. While I acknowledge that discussing negative or inconclusive findings can be challenging, I encourage the authors to review and strengthen this section.
For example, why was SWV used as a surrogate for elastic modulus when you had tendon stiffness and dimensions (i.e. could measure directly)? Have other studies found changes in these properties after training, and what method did they use? Are your results comparable or not?

Reviewer 2 ·

Basic reporting

- The English should be improved to ensure an international audience can understand your text clearly. Some examples where the language could be improved include lines 37, 40, 58, 80, 97, 115, and 169. Issues such as typographical errors (e.g., incorrect apostrophes), word choice (e.g., “comparably” instead of “comparatively”), and punctuation (e.g., wrong use of “ñ”) affect readability and clarity. I suggest you have a colleague proficient in English and familiar with the subject matter review your manuscript or consider using a professional language editing service to enhance the quality and precision of the language.

Experimental design

- The study recruited 39 participants, split into OC (n=20) and non-OC (n=19) groups (lines 87-89). Was a power analysis conducted to ensure sufficient sensitivity to detect differences in tendon stiffness and CSA changes? Given the small sample size, reporting confidence intervals for key outcomes would provide a clearer picture of result reliability.

Validity of the findings

- Lines 67-70 highlight the lack of detailed reporting on OC type and dosage, a standard limitation noted in the literature (Burrows & Peters, 2007). Given that different progestins vary in androgenicity, a detailed breakdown of participant-specific OC formulations would enhance the study’s generalizability and allow for subgroup analysis of different hormonal potencies.

- The authors mention discrepancies in tendon stiffness findings (lines 55-56). Bryant et al. (2008) and Morse et al. (2013) reported increased strain, while other studies found no differences in tendon stiffness or CSA. A discussion integrating biomechanical theories or highlighting differences in study populations (e.g., athletic vs. non-athletic) could contextualize these mixed results.

- Lines 57-59 mention discrepancies in previous findings regarding the influence of OC on tendon properties. However, discussing these conflicting results could be expanded to clarify how this study addresses the limitations or inconsistencies.

- Lines 59-70 briefly describe limitations in hormonal phase control and OC formulation reporting. Consider providing a more structured breakdown of these issues and how they justify the current research.

- Thank you for presenting detailed data and comprehensive results in your manuscript. The dropout and exclusion criteria are explained, but further clarification on the potential implications of age differences between groups (3-year age gap) on the outcomes is needed. This factor may affect interpretation and should be considered more explicitly in the discussion.

- The manuscript demonstrates rigorous statistical testing; however, key results (e.g., interaction effects) could be summarized more effectively using confidence intervals and effect sizes alongside p-values to convey the magnitude of differences.

- I appreciate your study’s thoughtful design and comprehensive exploration examining the effects of oral contraceptive (OC) use on patellar tendon (PT) adaptations to resistance training. The study benefits from a robust dataset, including hormonal profiles, tendon morphological properties, and biomechanical variables. The longitudinal design following participants across multiple menstrual or pill cycles provides valuable insight into the temporal nature of tendon adaptations. If there is a weakness, it is in the primary conclusion that OC use does not significantly inhibit tendon hypertrophy or biomechanical adaptations is well-articulated. However, the potential impact of statistical power and sensitivity on detecting subtle effects should be elaborated further. Including an analysis of the study’s power to detect interaction effects could provide context for non-significant results.

Annotated reviews are not available for download in order to protect the identity of reviewers who chose to remain anonymous.

Reviewer 3 ·

Basic reporting

Meets standards - comments contained within PDF

Experimental design

Meets standards - Comments contained within PDF

Validity of the findings

Meets standards - comments contained within PDF

Additional comments

n/a

Annotated reviews are not available for download in order to protect the identity of reviewers who chose to remain anonymous.

---

## Round 0.2 · Minor Revisions

Please address the remaining reviewer comment.

Reviewer 1 ·

Basic reporting

Thank you - I am satisfied with the manuscript changes.

Experimental design

Thank you - I am satisfied with the manuscript changes.

Validity of the findings

Thank you - I am satisfied with the manuscript changes.

Additional comments

I have one minor comment remaining, related to the ovulation of participants. I will upload this comment separately.

Annotated reviews are not available for download in order to protect the identity of reviewers who chose to remain anonymous.

Reviewer 3 ·

Basic reporting

Reporting meets publication standards.

Experimental design

The article meets publication standards

Validity of the findings

The article meets publication standards

Additional comments

The authors have provided a comprehensive rebuttal to each of the points I requested be addressed from the original review, and to a satisfactory level. There are limitations to the work submitted, but the authors have taken the correct steps to make these transparent throughout the manuscript and addressed them in a reasonable manner.

---

## Round 0.3 · accepted · Accept

I am pleased to inform you that all changes have been approved by the reviewer and the manuscript is ready for the next steps for publication

Reviewer 1 ·

Basic reporting

I am happy with the author responses.

Experimental design

I am happy with the author responses.

Validity of the findings

I am happy with the author responses.